# WHERE ARE THE BOTTLENECKS IN LONG-TAILED CLASSIFICATION?

## ABSTRACT

A commonly held belief in deep-learning based long-tailed classification is that the representations learned from long-tailed data are "good enough" and the performance bottleneck is the classification head atop the representation learner. We design experiments to investigate this folk wisdom, and find that representations learned from long-tailed data distributions substantially differ from the representations learned from "normal" data distributions. We show that the long-tailed representations are volatile and brittle with respect to the true data distribution. Compared to the representations learned from the true, balanced distributions, long-tailed representations fail to localize tail classes and display vastly worse inter-class separation and intra-class compactness when unseen samples from the true data distribution are embedded into the feature space. We provide an explanation for why data augmentation helps long-tailed classification despite leaving the dataset imbalance unchanged — it promotes inter-class separation, intra-class compactness, and improves localization of tail classes w.r.t to the true data distribution.

## 1 INTRODUCTION

Long-tailed distributions are those in which samples from a small number of head (majority) classes vastly outnumber the samples for a large number of tail (minority) classes. Such distributions can occur naturally, such as rare diseases in medical contexts or minority ethnic groups in face recognition. Learning from long-tailed data is challenging because it combines three problems: imbalanced learning across the head/body/tail, few-shot learning on the tail classes, and a label distribution shift at test time (long-tailed training set, balanced test set).

In practice, most research focuses on the data imbalance or label distribution shift problem, which reflects a communal belief that the bottleneck in long-tailed classification lies in the classifier rather than the representation — the representation is though to be "good enough". We question this belief and provide evidence against it through a series of experiments. First, we show that the representation is *not* good enough (§1.1) and may be a larger bottleneck than the classifier. We show that the long-tailed representations and "normal" representations have substantial differences in their second moments (§2.1), and that long-tailed representations have significantly worse inter-class separation and intra-class compactness (§2.2), and poorly localize tail classes in feature space (§8). Finally, we explain the reason why data augmentation boosts performance in long-tailed learning (Zhong et al.; Zhang et al., 2021b) (§1.2)) despite having no effect on the imbalance factor, showing that it significantly improves localization of classes (§8), and confers robustness to distribution shift by improving inter-class separation and inter-class compactness (§2.2.

### 1.1 IS THE REPRESENTATION OR CLASSIFIER THE BOTTLENECK?

There is a tension between representation learning and classifier learning in long-tailed classification, due to their opposite locations in the bias-variance trade-off spectrum. Representation learning is thought to suffer more from data scarcity rather than data imbalance (Yang & Xu, 2020), while classifier learning is though to suffer more from data imbalance. Decoupled training (Zhong et al.; Kang et al., 2020) methods attempt to address this tension by training the classifier and representation separately (left side of Fig 1). It is a common belief that the representations are "good enough"

**Standard Decoupled Training (cRT)**  **Experimental Configuration**

Figure 1: The standard classifier retraining (cRT) setup on the left, and the experimental configuration we use on the right. Instead of resampling, we swap datasets between stages to simulate an "ideal" classifier or an "ideal" representation.

| Dataset | Accuracy | | | | Backbone |
| | $\mathcal{D}_{LT} \to \mathcal{D}_{LT}$ | $\mathcal{D}_{LT} \to \mathcal{D}^\star$ | $\mathcal{D}^\star \to \mathcal{D}_{LT}$ | $\mathcal{D}^\star \to \mathcal{D}^\star$ | |
|---|---|---|---|---|---|
| CIFAR-10 LT | 0.7212 | 0.8261 | 0.9106 | 0.9302 | ResNet-32 |
| CIFAR-100 LT | 0.409 | 0.5157 | 0.662 | 0.7109 | ResNet-32 |
| ImageNet-LT | 0.3531 | 0.4586 | 0.5383 | 0.6393 | ResNet-10 |

Table 1: Models with classifiers trained on long-tailed distributions but representations trained on the true distribution are less-biased than models with representations trained on the long-tailed distribution and classifiers trained on the true data distribution. A high quality representation can resist a highly biased classifier.

and the bottleneck is the classifier (Zhang et al., 2021a; Yang & Xu, 2020; Kang et al., 2020), thus many works focus on interventions in the classifier learning phase (Menon et al., 2020).

We test this belief by carrying out an altered form of decoupled training (right side of Fig 1). The three most commonly used long-tailed datasets — CIFAR-10/100 LT and ImageNet-LT (Liu et al., 2019) and ImageNet-LT — all have larger corresponding balanced datasets, namely CIFAR-10/100 (Krizhevsky) and ImageNet-1k (Deng et al., 2009). Let $\mathcal{D}_{LT}$ denote the long-tailed version of a dataset, and $\mathcal{D}_\star$ denote the normal, balanced version. To assess the relative impact of the representation and classifier, we apply decoupled training *without* resampling. Instead of resampling, we swap between the $\mathcal{D}_{LT}$ and $\mathcal{D}^\star$ versions of the dataset between Stage 1 and Stage 2. If the classifier is truly the bottleneck, we expect to see that a representation trained on $\mathcal{D}^\star$ with a classifier retrained on $\mathcal{D}_{LT}$ should perform worse on the balanced test set than a representation trained on $\mathcal{D}_{LT}$ with a classifier retrained on $\mathcal{D}^\star$.

We find that this is not true (Table 1). As expected, the $\mathcal{D}^\star \to D^\star$ model (with a representation and classifier trained on the true, balanced dataset), performs the best. However, across all three datasets, the $\mathcal{D}^\star \to D_{LT}$ model outperforms the $\mathcal{D}_{LT} \to \mathcal{D}^\star$ model by a substantial margin. This indicates that contrary to popular belief, the bias in the representation dominates the bias in the classifier, and not the other way around.

## 1.2 WHY DOES DATA AUGMENTATION HELP?

At a first glance, it is unclear why class-agnostic input data augmentation should improve long-tailed classification. Data augmentation is incapable of addressing data imbalance — it leaves the imbal-

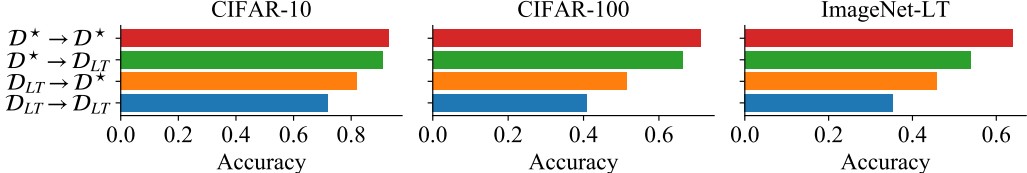

Figure 2: Table 1 in graphic form, showing that a biased representation is very difficult to overcome, but a biased classifier can be mitigated if the representation is unbiased.

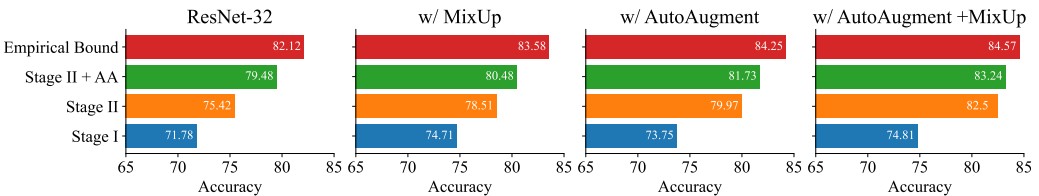

Figure 3: Data augmentation improves fair classification performance in long-tailed settings (CIFAR-10 with an imbalance factor of 100)

ance factor of a dataset unchanged. Nevertheless, input data augmentation such as MixUp (Zhang et al., 2018), Manifold Mixup (Verma et al., 2019) and AutoAug (Cubuk et al., 2019) have proven to be highly effective for reducing bias in long-tailed classification (Zhong et al.; Zhang et al., 2021b; Tan et al., 2020). When combined with decoupled training (Kang et al., 2020), data augmentation results in even larger increases (Fig 3). However, the mechanism by which data augmentation improves performance and reduces bias is not known. It cannot address data imbalance, so data augmentation must be affecting another quality of the representation.

### 1.3   WHY ISN'T REBALANCING ENOUGH?

Resampling and reweighting strategies have been an essential part of long-tailed classification. However, as first observed by Zhang et al. (2021a), rebalancing does not seem to be enough. Following the methodology of Zhang et al. (2021a), we construct an empirical classifier bound by first training a representation on the long-tailed $\mathcal{D}_{LT}$ dataset, in this case CIFAR-10 with an imbalance factor of 100. Next, we follow the standard decoupled training startegy for cRT (Kang et al., 2020) and retrain the classifier on version of $\mathcal{D}_{LT}$ that has been balanced by class-aware resampling Kang et al. (2020) after freezing the representation. To obtain the empirical classifier performance bound, we then take the frozen representations from the first stage and train a new classifier on the true dataset $\mathcal{D}^\star$. The performance of the classifier trained on $\mathcal{D}^\star$ can be viewed as an empirical upper bound on the performance obtainable through any resampling strategy, and is the ideal performance to aim for. There is a consistent gap between the class-aware resampling strategy and the upper bound (Fig 3), suggesting that the resampling strategy alone leaves performance on the table. Moreover, applying data augmentation again in the second stage boosts the performance even further, which is suprising in light of the fact that the classifier is primarily thought to suffer from imbalance — so why should applying data augmentation in the classifier learning phase have such a big impact on performance?

## 2   EXPERIMENTS

In the previous section, we showed results that provoke three questions.

1. Why are the representations learned from the long-tailed data so much poorer than the representations learned from the true dataset?

2. Why does data augmentation help in the representation learning phase of decoupled-training for long-tailed classification?

3. Why isn't rebalancing enough to fix the linear classifier's bias?

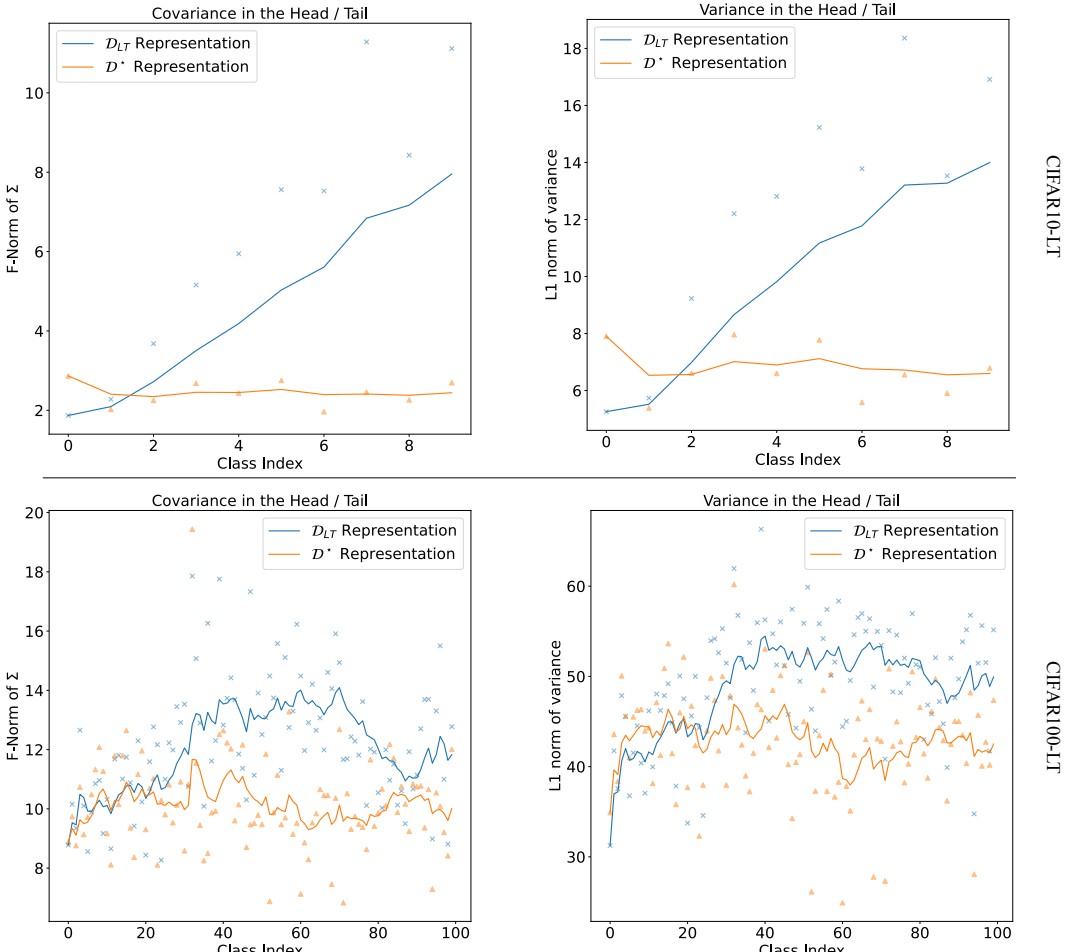

Figure 4: The second moments of features learned from long-tailed distributions are substantially different from those of representations learned from the true distribution. Variance for the tail classes is higher, and the covariance matrices have larger norms for the body and the tail.

We provide answers to these questions by analyzing the differences between the representations learned from the long-tailed data, the representations learned from the true dataset, and representations learned from the long-tailed data with augmentation.

## 2.1 REPRESENTATIONS LEARNED FROM LONG-TAILED DATA HAVE DIFFERENT 2ND MOMENTS

We train a ResNet-32 on a true, balanced dataset $\mathcal{D}^\star$ and a ResNet-32 on a long-tailed dataset $\mathcal{D}_{LT}$. We then extract features from both models for the entire dataset ($\mathcal{D}^\star$), to reflect the true data distribution. We use CIFAR10/100-LT with an imbalance factor of 100.

The representations learned from long-tailed data differ from the representations learned from the true dataset in two significant ways. First, there are notable differences in the variance and co-variance of the classes that belonged to the head or tail in the long-tailed dataset within the $\mathcal{D}_{LT}$ representation, but not in the $\mathcal{D}^\star$ representations (Fig 4). Specifically, the variance of the features belonging to the tail and body classes (the L1-norm of the covariance matrix diagonal) diverges

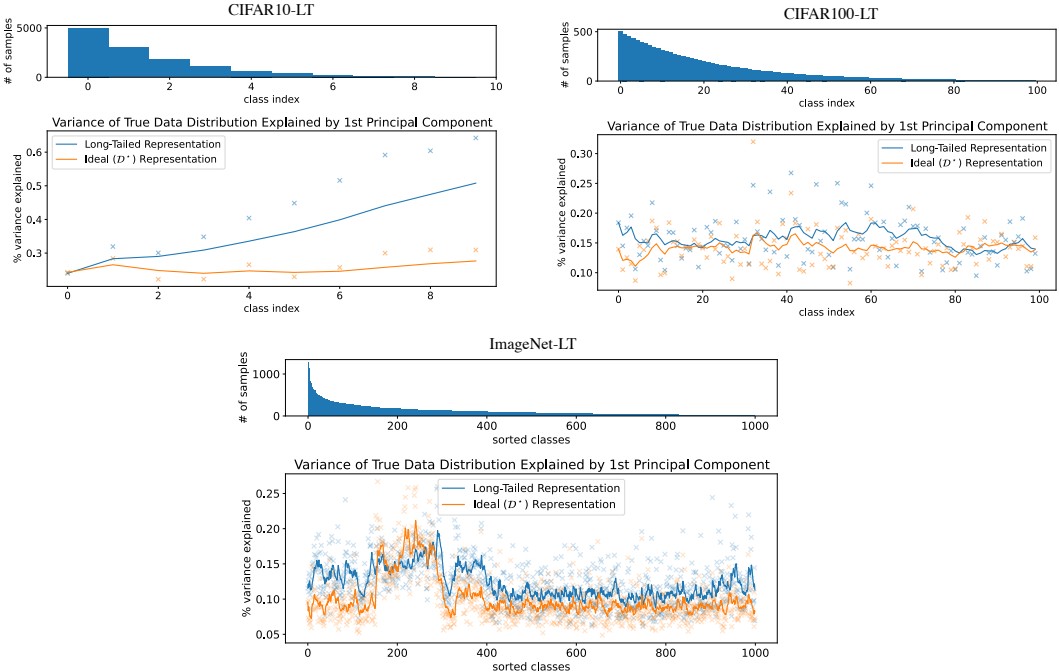

Figure 5: The feature space of long-tailed representations is biased — their variance is concentrated in fewer dimensions than those of the representations trained on the true class distributions. This can be seen by the relatively higher proportion of variance explained by the first principal component.

substantially from the variance of the features belonging to the head classes. This is not only true of the variance, but of the covariance. The Frobenius norm of the sample covariance matrix computed from the features of a class is also different between the $\mathcal{D}^\star$ and $D_{LT}$ learners. For the learner trained on $\mathcal{D}^\star$, there are no significant differences in variance or covariance between the head, tail, and body, and differences are dominated by noise among the classes. For the learner trained on $\mathcal{D}_{LT}$, the variance of the head is systemically lower than the variance of the head, and this is also true of the covariance.

Next, we examine the proportion of variance explained by the first principal component of the data matrix for each class. Concretely, we take all feature vectors belonging to the class from the complete dataset $\mathcal{D}^\star$, and compute the first principal component of the data matrix of features for the class. We then plot the proportion of variance explained by the first principal component of each class (Fig. 5), thus obtaining an estimate of the relative complexity of the subspace each class resides in. Across all three datasets, we see the same result: the first principal component explains more of the variance in the representations learned from $\mathcal{D}_{LT}$ than the representations learned from $\mathcal{D}^\star$.

Intuitively, these results suggest that the representations learned from long-tailed data are substantially different than the representations learned from the true, balanced dataset. The differences are most marked for tail and body classes, and least for the head classes. Geometrically, the long-tailed representations appear to occupy a less complex subspace than the $\mathcal{D}^\star$ representations, yet exhibit greater variance, suggesting that the feature space is biased — fewer dimensions in the long-tailed feature space are relevant to all classes, and thus the variance concentrates in fewer dimensions of the feature space.

## 2.2 REPRESENTATIONS LEARNED FROM LONG-TAILED DATA ARE MORE DIFFUSE AND LESS SEPARABLE

We now examine the representations from the angular perspective. Recall that the score for the class $k$ logit in a standard linear classifier for a feature vector $\mathbf{x}$

$$\mathbf{w}_k \cdot \mathbf{x} = ||\mathbf{w}_k|| \times ||\mathbf{x}|| \times \cos\theta \qquad (1)$$

**Differences in intra-class compactness between representations**

Figure 6: More unseen samples from the true data distribution fall outside the angular maximum of the tail class distributions for long-tailed representations, than for representations trained on the true distributions. Augmentation can mitigate this effect.

is a function of the angle $\theta$ between $\mathbf{x}$ and $\mathbf{w}_k$ when $||\mathbf{w}_k|| = ||\mathbf{x}||$ In practice, the norm $||\mathbf{w}_k||$ of the logits of a class $k$ are strongly correlated with the cardinality of the class (Menon et al., 2020; Kang et al., 2020; Zhong et al.) — tail classes have smaller norms and vice versa, reflecting the class priors. However, decoupled training roughly equalizes the logit norms Kang et al. (2020), allowing us to ignore the $||\mathbf{w}_k|| \times ||\mathbf{x}||$ term and focus only on the angle $\theta$. In this setting, the classifier reduces to an angular classifier, and a sample $\mathbf{x}$ will be assigned to the logit $w_k$ to which it has the smallest angular distance. Thus, we proceed to an examination of the angular distribution created by the features of the long-tailed representation learner and the $\mathcal{D}^\star$ representation learner.

The representations of the tail classes learned by the long-tailed representation learner are significantly less compact than the representations learned by the $\mathcal{D}^\star$ representation learner with respect to the true data distribution (Fig 6). While the long-tailed per-class representations are compact w.r.t to the long-tailed distribution, when the unseen samples making up the true data distribution of $\mathcal{D}^\star$ are added, the angular spread significantly increases for the tail classes. This means that any classifier boundaries learned using only the samples available in the $\mathcal{D}_{LT}$ distribution will fail to generalize, as the majority of unseen samples for a tail will escape the classification boundaries. Interestingly, augmentation appears to significantly reduce the compactness gap between the long-tailed and true data distribution for the tail classes. This may partly explain the success of data augmentation in long-tailed classification. However, data augmentation (MixUp) also significantly increases the angular spread of *all* classes, in contrast to the representation trained on $\mathcal{D}^\star$, which addresses the distribution shift without compromising the angular spread of the classes.

The representations of the tail classes learned by the long-tailed representation learner also display worse inter-class separation with respect to the true data distribution (Fig 7). The angular distance to the nearest *incorrect* class center significantly shrinks when the unseen samples from the true data distribution are considered. Again, augmentation makes a significant difference: the angular distribution shift of representations trained on the long-tailed dataset with augmentation is far smaller than the angular distribution shift of representations trained on the long-tailed dataset without augmentation.

Thus, two major differences between the representations learned from long-tailed data and representations learned from the $\mathcal{D}^\star$ data is the effect of distribution shift on intra-class compactness and

inter-class separation. When unseen samples from the true data distribution are added, the angular distributions of the tail classes balloon for the representation trained on long-tailed data. The poor intra-class compactness also leads to poor inter-class separation w.r.t to the true data distribution, as the expansion of angular distributions results in each class eating into the margin between it's nearest neighbor. This means that classifier boundaries drawn on the long-tailed distribution will likely be optimistic and fail to generalize due to unseen samples from the true distribution mostly escaping the convex polytope of the long-tailed classifier boundaries. However, data augmentation seems to help, because the representations learned from long-tailed data with data augmentation exhibit much better inter-class separation and intra-class compactness under distribution shift.

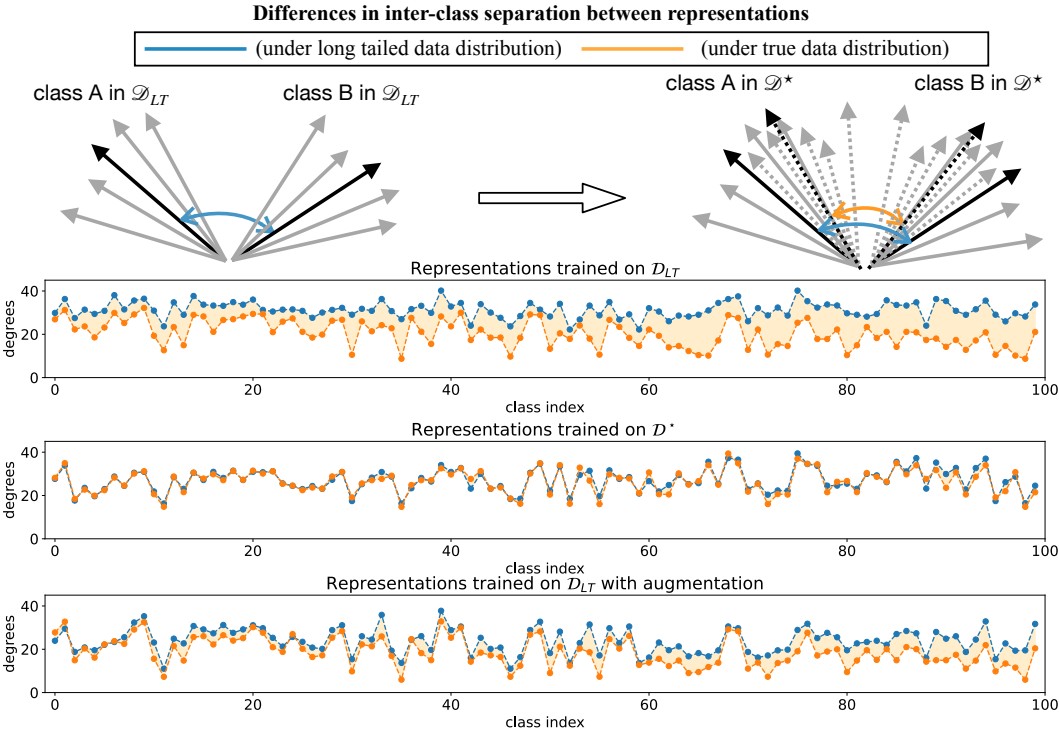

Figure 7: When unseen samples from the true data distribution are embedded into the long-tailed feature space, inter-class separation significantly drops. This is not true for the long-tailed distribution, which sees relatively small shifts in inter-class separation. Augmentation mitigates this effect, and augmented representations experience a small change in inter-class separation.

### 2.3 REPRESENTATIONS LEARNED FROM LONG-TAILED DATA POORLY LOCALIZE TAIL CLASSES

Intra-class compactness and inter-class separation have known fixes: apply a margin. However, margins will only be minimally helpful if the class centers are very poorly localized, which we now show occurs for long-tailed representation learnes. Proceeding with the experimental setup of the previous sections, we examine the angular shift induced in the class centers by the distribution shift between the long-tailed distribution and the true $\mathcal{D}^\star$ distribution. Specifically, let $\mu_k$ be the center of class $k$ with respect to the long-tailed dataset $\mathcal{D}_{LT}$, and let $\mu'_k$ be the center of class k with respect to the true data distribution (the $\mathcal{D}^\star$ dataset). We show that for the tail classes , the angular change $\theta_{\Delta_k} = \mu_k - \mu'_k$ in the location of the class center for class $k$ induced by knowledge of the true data distribution is substantial for the long-tailed learner, nearly equivalent to the angular width of the class itself. Knowledge of the true data distribution also significantly expands the variance of the angular distribution proportional to the inverse class frequency for the tail classes in the long-tailed representation learner.

Again, augmentation has an ameliorating effect and helps to repair this problem in the long-tailed representation. The angular shift in the class centers of $\mathcal{D}_{LT}$ induced by adding the unseen sam-

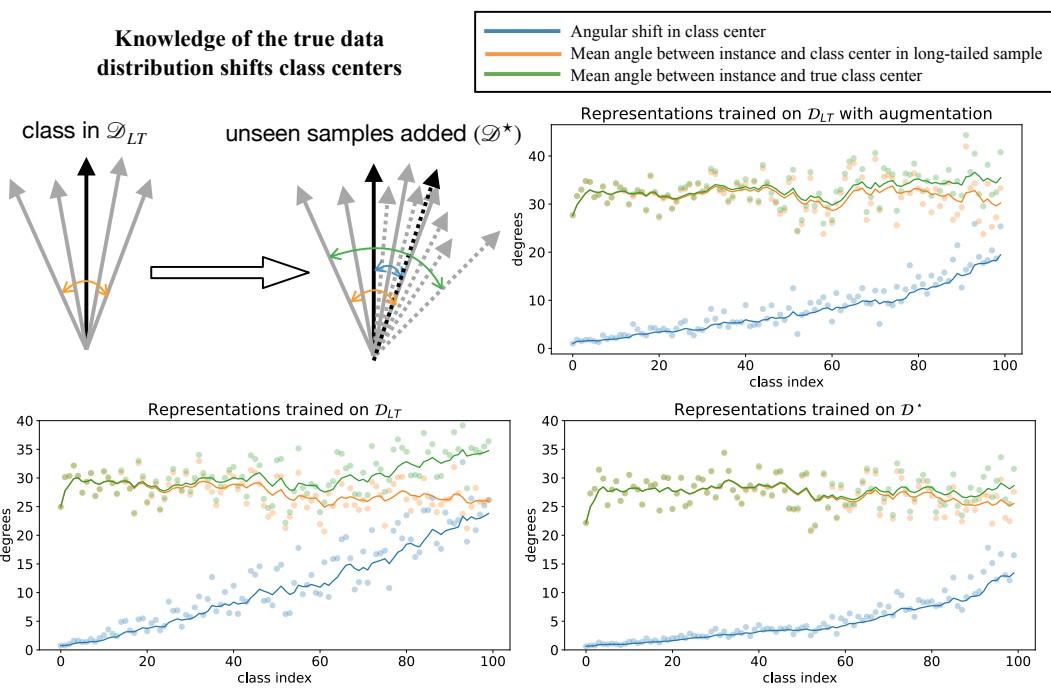

Figure 8: Tail classes are poorly localized w.r.t to the true data distribution for long-tailed representations. The class centers experience a large angular shift, nearly equal to the width of the classes for classes in the tail.

ples of $\mathcal{D}^{\star}$ to $\mathcal{D}_{LT}$ is significantly lower for the augmentation representation than the unaugmented representation. In the augmented representation, the increase in angular variance induced by knowledge of the true data distribution is also lower, meaning the augmented representations both better localize the class centers and are more resistant to distribution shift and thus likelier to generalize.

## 3 IMPLEMENTATION DETAILS

For experiments, we create artificially long-tailed versions of the CIFAR-10/100 (Krizhevsky) datasets by following the procedure of Cui et al.. Specifically, the number of samples $n_i'$ in the $i$-th class is set to be $n_i' = n_i\mu^i$, where $n_i$ is the original number of samples in the class, and $\mu \in [0,1]$ is used to control the imbalance factor, defined as the count of samples in the largest class divided by the count of samples in the smallest class. For all experiments, we use a long-tailed version of CIFAR-100/10 with imbalance factor of 100. Following Zhang et al. (2021b); Cui et al., we train a ResNet-32 He et al. (2015) for 200 epochs using the SGD optimizer with an initial learning rate of 0.01, with 5 epochs of warm start Goyal et al. (2018), a weight decay of $2e - 4$, momentum of 0.9, and divide the learning rate by 100 at the 160th and 180th epochs using a batch size of 128. For ImageNet-LT, we follow the setting of Liu et al. (2019) and train a ResNet-10 for 100 epochs with a weight decay of $1 \times 10^{-4}$ using SGD with a momentum of 0.9 and an intital learning rate of 0.2 with a batch size of 512. The learning rate is divided by 10 at epochs 30, 60, and 80. For the classifier learning phase, we follow (Zhong et al.) and use a cosine learning rate scheduler with a batch size of 512 for 10 epochs. For both CIFAR10/100-LT and ImageNet LT, we use an initial learning rate of 0.05 and a weight decay of $2e - 4$ in the classifier learning phase. We found that the $\alpha$ value for MixUp had minimal effect over the values $0.6, 0.8, 1.0$, and fixed it at $1.0$.

## 4 RELATED WORK

Feldman & Zhang (2020) develop an intriguing theory for long-tailed learning in particular, in which they claim that the primary obstacle to learning from natural datasets is not label noise, but insuf-

ficent samples for rare and atypical examples. They provide experimental evidence in Feldman (2021). In the case of decoupled learning, Zhou et al. (2020) makes an attempt to identify *why* decoupled learning works. They conclude that class-level rebalancing strategies harm the representation learning. Specifically, class-level rebalancing results in better separability of classes, but causes the within-class distribution of samples to become more diffuse and spread, instead of compact. Kornblith et al. (2020) notes that all loss functions which improve over the standard softmax do so by producing greater class separation in the final representations. Starting with CenterFace Wen et al. (2016) and continuing on to SphereFace Liu et al. (2017) and ArcFace Deng et al. (2019), large-margin loss functions have driven significant performance improvements in face recognition. To achieve perfect accuracy when metric learning, it is necessary that the maximal intra-class distance is smaller than the minimal inter-class distance to allow a nearest-neighbor classifier to work correctly. This requires the metric to push class centers apart, while compacting the "feature clouds" of each class. The results of Zhou et al. (2020) suggests that only separability is enhanced by class rebalancing, while intra-class compactness is harmed. Liu et al. (2020) notice that in the long-tailed setting under a cosine classifier, the angles made by the head classes with respect to the class center exhibit larger variation than the angles made by the tail classes. Khan et al. (2019) show that Bayesian uncertainty estimates (equivalent to the variance of a Gaussian process) are negatively correlated with the number of samples — tail-classes have higher uncertainty (variance) estimates. Yang et al. (2018) point out that CNNs trained with this loss only learn a linearly separable space, and therefore have no reason to be intra-class compact. Both Zhang et al. (2021b) and Zhong et al. test MixUp Zhang et al. (2018) and find that it results in substantial performance gains. Yang & Xu (2020) leverage unlabeled additional data with semi-supervised learning and a pseudo labeling loss to train a classifier, showing that semi/self-supervised representations may have advantages supervised representations do not.

## 5    CONCLUSION

We probe the fundamental differences between representations learned from long-tailed data and representations learned from the true data distribution, and use those differences to explain why data augmentation improves long-tailed classification. Contrary to popular wisdom, we show that representation has greater potential to be the bottleneck in long-tailed classification than the classifier. Our characterization illustrates the primary differences between long-tailed representations and those learned from the true class distributions: biased feature space, less separable and more diffuse classes, and poor localization of tail classes. The results suggest that the primary problem in long-tailed classification may in fact be the few-shot learning problem on the tail classes, rather than a data imbalance problem.

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

## A APPENDIX

You may include other additional sections here.

