# OpenReview forum: "Where is the bottleneck in long-tailed classification?"
_ICLR.cc/2022/Conference — ICLR 2022 Submitted_

### Official Review · Reviewer_B7am · 2021-10-19

**Correctness:** 2
**Technical Novelty And Significance:** 2
**Empirical Novelty And Significance:** 2
**Recommendation:** 3
**Confidence:** 5

**Main Review:**

Positive points:
1. This paper seeks to empirically investigate the importance of representation learning, which may provide a new understanding of deep long-tailed learning to the community.


2. This work shows the effectiveness of intra-class compactness and inter-class separation on long-tailed representation learning.


3. This work also analyzes the influence of data augmentation on long-tailed representation learning, which provides a better understanding of data augmentation in deep long-tailed learning.

Negative points:
1. This paper mentioned that "a commonly held belief in deep long-tailed classification is that performance bottleneck is the classification head atop the representation learner". However, such a belief may not be common. Note that many recent long-tailed learning studies focus on improving representation learning [1], e.g., KCL [2], Hybrid [3], PaCo[4] and DRO-LT [5].  Moreover, in the conclusion, this paper stated that "the results suggest that the primary problem in long-tailed classification may in fact be the few-shot learning problem on the tail classes, rather than the imbalance problem". However, this argument is too strong and the obtained results cannot support it. It would be better if the authors had written all the arguments more rigorously and verified them more completely.


2. As mentioned by the above question, there are many representation learning-based long-tailed studies, e.g., [1-5]. Therefore, it would be better if authors can review mire representation learning based long-tailed learning methods in related work.


3. The vital problem in this paper is the used balanced set: i.e., CIFAR-10/100 and ImageNet-1K. Please note that the data number of  CIFAR-10/100 or ImageNet-1K is much more than their long-tailed variants, i.e., CIFAR-LT and ImageNet-LT. Considering using more training samples will lead to significant improvement in representation learning and model performance, most empirical comparisons in this paper (especially Table 1 and Figure 2) are unfair and the corresponding arguments are unpersuasive. The experiments would have been more persuasive, if the balanced training set is a variant of the long-tailed training set with a similar total data number but each class has the same/similar data number, like [1,2]. For example, a balanced set of ImageNet-LT can be obtained at https://github.com/Vanint/Awesome-LongTailed-Learning/tree/main/resources/data_txt/ImageNet_LT.


4. Please discuss more of Figure 5. On ImageNet-LT and CIFAR100-LT, the variance trends of long-tailed representations and ideal representations are quite consistent. Such observations seem different from the conclusion of Sec 2.1. Since I am confused about the results, I guess other readers may also do so. Therefore, I suggest the authors explain them more.




Minor suggestions:
1. Figure 3 is not clear enough. It would be better if the authors can explain it more in the captions. Moreover, what are "degrees" in Fig.6-8? Please make them more clear.

2. In Lines 4-5 of page 2, ImageNet-LT appears twice. In Line 1 of page 6, there should be a "full stop" before In practice.


References:

[1] Deep long-tailed learning: A survey. ArXiv, 2021.

[2] Exploring balanced feature spaces for representation learning. In ICLR, 2021.

[3] Contrastive learning based hybrid networks for long-tailed image classification. In CVPR, 2021.

[4] Parametric contrastive learning. In ICCV, 2021.

[5] Distributional robustness loss for long-tail learning. In ICCV, 2021.



**Summary Of The Paper:**

This paper seeks to study what is the bottleneck in long-tailed learning. Based on extensive experiments, the authors propose that representation learning is the bottleneck in long-tailed classification. Also, this paper analyzes representation learning from the perspectives of intra-class compactness and inter-class separation, as well as the influence of data mixup on long-tailed representation learning.

**Summary Of The Review:**

Overall, I like the goal of this paper, i.e., analyzing the bottleneck of long-tailed learning. However, I cannot champion this paper since the data number of the used balanced set is much larger than the long-tailed set, which makes the empirical comparisons unfair and the corresponding finding unpersuasive. Moreover, the arguments in this paper should be written more rigorously. I am glad to see the response of the authors.

---

> ### Author Response · Authors · 2021-11-23
> **Additional experiments to address flaws**
>
> Thank you for the great feedback and enthusiasm! We shall add these papers to the literature review section to cover more perspectives on representation learning in the long-tailed setting. It is indeed true that representation learning is becoming more common, however, it is also true that the dominant paradigm thus far has been addressing imbalance. We wish to encourage work on representation learning in the long-tailed setting by pointing out its importance.
>
> We have conducted additional experiments by adding a third category of dataset, $\bar{\mathcal{D}}$. In $\bar{\mathcal{D}}$, the number of samples is the same as $\mathcal{D}_{LT}$, with the only difference being that each class has the same number of samples — there are no more few shot classes.
>
> **Additional Experiments with new balanced set**
>
> CIFAR10 | $\bar{\mathcal{D}} \rightarrow \mathcal{D}^\star$ (0.871) >  $\\mathcal{D_{LT}} \rightarrow \mathcal{D}^\star$ (0.826)
>
> CIFAR100 | $\bar{\mathcal{D}} \rightarrow \mathcal{D}^\star$ (0.55) >  $\\mathcal{D_{LT}} \rightarrow \mathcal{D}^\star$ (0.515)
>
> ImageNet | $\bar{\mathcal{D}} \rightarrow \mathcal{D}^\star$ (0.478) >  $\\mathcal{D_{LT}} \rightarrow \mathcal{D}^\star$ (0.458)
>
> In all cases, the representation learned by the balanced dataset **with the name number of samples as the long-tailed dataset** outperforms the representation learned by the long-tailed dataset.
>
> **Meaning of Fig 5**
> The intention of Fig 5 is to show an aspect of the differences between the ideal representation (orange line) and the long-tailed distribution (blue line). It can be seen that the blue line is consistently higher than the orange line for most classes, showing that the variance captured by the first component is higher for the long tailed representation than the ideal representation. This is, to our knowledge, a novel observation. We are showing that one of the differences between long-tailed representations and the ideal representation is that the variance is concentrated in fewer dimensions.
>
> **Strength of claim** It is difficult to weaken the claim "the results suggest that the primary problem in long-tailed classification may in fact be the few-shot learning problem on the tail classes, rather than the imbalance problem" much further. We are only saying that the few-shot problem may be a larger contributor to the performance degradation than the imbalance, and the experiments do suggest this. We are not saying that imbalance is not a factor, but that the data scarcity is a *bigger problem* than the imbalance. We hope the new experiments can support this claim.
>
> **Other points** We want to point out that another goal of our paper is to study the differences between long-tailed representations and ideal representations learned from the full data distribution. Our belief is that if we can point out these differences, other researchers may be able to use them to guide the development of new methods.

---

### Official Review · Reviewer_eREq · 2021-11-01

**Correctness:** 4
**Technical Novelty And Significance:** 3
**Empirical Novelty And Significance:** 4
**Recommendation:** 8
**Confidence:** 5

**Main Review:**

I really enjoyed reading the paper. It has a very clear direction from the beginning with good experiments to back it up. The writing is clear as well. I believe long-tailed classification is an interesting problem with clear real-world applications, so studying it in-depth is necessary for the community.

Overall, I don't see any major drawbacks or shortcomings as the experiments and ablations combined with the analysis are solid. That said I have few questions:

The difference in representation between D* and D_{LT} is clearly visible. However, apart from difference in the shape of distribution (long-tail vs balanced) there is also difference in the amount of data between those two which might play an important role, especially when considering learned representations. It would be good to see what is the difference as well between two datasets that have equal number of examples, but different distributions: normal ad LT. Since otherwise the authors conclusion might raise a question.

Additionally,  Yin et al. [1] performed a related analysis of classifier magnitude difference which was depending on the distribution of classes. The authors analysis seems deeper here, however it would be good to address any similarities/differences. On top of that, few methods [2, 3] used adversarial examples in order to modify the learned representations instead of the classifiers in LT task. It would be good to see what the authors think about such direction and what impact on the feature space and measured statistics it would have.

And finally, apart from the analysis, what are the conclusions here for the future researchers - any thoughts on proposed directions/approaches that could originate from the performed analysis?

[1] Yin, Xi, et al. "Feature transfer learning for face recognition with under-represented data." Proceedings of the IEEE/CVF Conference on Computer Vision and Pattern Recognition. 2019.

[2] Kim, Jaehyung, Jongheon Jeong, and Jinwoo Shin. "M2m: Imbalanced classification via major-to-minor translation." Proceedings of the IEEE/CVF Conference on Computer Vision and Pattern Recognition. 2020.

[3] Kozerawski, Jedrzej, et al. "BLT: Balancing Long-Tailed Datasets with Adversarially-Perturbed Images." Proceedings of the Asian Conference on Computer Vision. 2020.

**Summary Of The Paper:**

The authors study the long-tail dataset problem in order to determine the true bottleneck for the task. After performing many ablations and experiments on 3 benchmark datasets they establish that contrary to common belief the bottleneck is in data representation rather than the classifier itself.

**Summary Of The Review:**

Good analytical paper on interesting subject of long-tailed classification. It would be good to see authors thoughts on impact of the amount of data in training, impact of adversarial augmentations, and proposed directions stemming from the analysis.

---

> ### Author Response · Authors · 2021-11-23
> **Further discussion**
>
> Thank you for the positive review and engagement with our submission!
>
> We have conducted additional experiments by adding a third category of dataset, $\bar{\mathcal{D}}$. In $\bar{\mathcal{D}}$, the number of samples is the same as $\mathcal{D}_{LT}$, with the only difference being that each class has the same number of samples — there are no more few shot classes.
>
> **Additional experiments with new balanced set**
>
> CIFAR10 | $\bar{\mathcal{D}} \rightarrow \mathcal{D}^\star$ (0.871) >  $\\mathcal{D_{LT}} \rightarrow \mathcal{D}^\star$ (0.826)
>
> CIFAR100 | $\bar{\mathcal{D}} \rightarrow \mathcal{D}^\star$ (0.55) >  $\\mathcal{D_{LT}} \rightarrow \mathcal{D}^\star$ (0.515)
>
> ImageNet | $\bar{\mathcal{D}} \rightarrow \mathcal{D}^\star$ (0.478) >  $\\mathcal{D_{LT}} \rightarrow \mathcal{D}^\star$ (0.458)
>
> In all cases, the representation learned by the balanced dataset **with the name number of samples as the long-tailed dataset** outperforms the representation learned by the long-tailed dataset.
>
> **Missing Related Work** These are very good suggestions, and we shall add them to the related works section. Thank you for pointing them out! The main difference between our analysis and the analysis of [1] is that we simplify the setting to depend on the angle only, so the logits all have the same magnitude. Since [1] shows that logit magnitude is correlated with prior class probability, our intuition is that the angular margin assumption is similar to enforcing a uniform prior on the class probability distribution. As for [2,3], these are intimately related. Adversarial perturbations are stronger forms of data augmentation, so we expect that they would exhibit similar effects to data augmentation, but stronger in magnitude.
>
> **Proposed Future Directions / Approaches** We see two possible approaches originating from this work. One focuses on explicitly enforcing desired properties such as intra-class compactness, inter-class separation, etc by building them into the loss function, as is done in large-margin methods. The other thread is *implicitly* promoting these properties by data augmentation. This includes adversarial perturbations, stronger forms of MixUp, or synthetic data augmentation. Furthermore, it would be interesting to see if these methods can be combined to yield compounding benefits, or if they do not work well with each other.
>
>
> [1] Yin, Xi, et al. "Feature transfer learning for face recognition with under-represented data." Proceedings of the IEEE/CVF Conference on Computer Vision and Pattern Recognition. 2019.
>
> [2] Kim, Jaehyung, Jongheon Jeong, and Jinwoo Shin. "M2m: Imbalanced classification via major-to-minor translation." Proceedings of the IEEE/CVF Conference on Computer Vision and Pattern Recognition. 2020.
>
> [3] Kozerawski, Jedrzej, et al. "BLT: Balancing Long-Tailed Datasets with Adversarially-Perturbed Images." Proceedings of the Asian Conference on Computer Vision. 2020.

---

### Official Review · Reviewer_jXk1 · 2021-11-03

**Correctness:** 2
**Technical Novelty And Significance:** 2
**Empirical Novelty And Significance:** 2
**Recommendation:** 3
**Confidence:** 4

**Details Of Ethics Concerns:**

I do not have any ethics concerns.

**Main Review:**

Strength:
1. The topic is interesting and the papers pose some unique observations after extensive empirical analysis and experiments
2. The paper defines several simple mathematical and statistical metrics to measure the differences between representations.

Weakness：
1. The posed questions are not well addressed
The paper shows some observations but did not provide a concrete and reasonable solution such that the long-tailed classification issue can be addressed. The useful insight from this paper is limited.

2. The empirical observations are not solid and rigorous
The paper only provides some simple metrics but did not explain why the metric is necessary and what's the high-level intuition. I did not get the motivation why the authors come up with these metrics to show the differences between representation. In addition, there is no rigorous mathematical proof or statistical analysis.

3. Lack of careful related work discussion
The related work section is hard to follow and the authors did not explain their contributions and differences from existing works.

4. The writing needs to be improved.
Many typos result in additional difficulties to read.   The citation format is not consistent. For example,  Cui et al.. in section 3 does not have a   year, and "He et al (2015)" followed by Resnet-32 should be "(He et al, 2015)".  The "google enough" and "Normal" in the abstract should be corrected.




**Summary Of The Paper:**

This paper poses an interesting and important question - where are the bottlenecks in long-tailed classification. The authors use empirical experiments to show their observations: (1) representation is more critical than classifier, (2) data augmentation is helpful.  Three datasets (CIFAR-10 LT, CIFAR-100 LT and ImageNet-LT) are employed to work with ResNet-32 and ResNet-10 models to demonstrate their observations.

**Summary Of The Review:**

An overall feeling is that the paper is an ongoing work and needs to be carefully written and improved. My recommendation is to reject it in the current form.

---

> ### Author Response · Authors · 2021-11-23
> **Intuition for metrics and differences from related works**
>
> Thanks for the comments, and showing interest in the paper!
>
> 1. "The paper shows some observations but did not provide a solution" **Our paper is not about developing a new technique, but rather about explaining phenomena**. We shed light on how learning from long-tailed data weakens representations. We focus on 3 metrics (inter-class separation, intra-class compactness, and distribution shift of tail classes w.r.t to the true data distribution) to quantitatively illustrate differences between long-tailed representations and ideal representations. Furthermore, we provide evidence that data scarcity is as much or more of a problem in long-tailed learned as imbalanced data. We also explain how and why data augmentation improves performance in long-tailed learning.
>
> 2. We did not invent these metrics. Some of these metrics (intra-class compactness and inter-class separation) are known in many fields, one example is large-margin methods in face recognition [1,2]. Angular shift has been explicitly discussed in the context long-tailed learning [3], while distribution shift has been a topic in few-shot learning [4]. We provide intuitive explanations of intra-class compactness after Eq.1 and in Fig 6. We also give a visual illustration of interclass separation (Fig. 7) and angular shift (Fig 8). These measures were chosen because (1) they describe geometric properties of representations (2) they are explicitly desirable properties in any representation for classification. Specifically, inter-class separation and inter-class compactness ensure that samples of the same class are packed closely together yet far from samples of other classes. Angular distribution shift describes how the sample mean and variance of the angle a collection of vectors make with a reference vector changes when the distribution of vectors changes.
>
> 3. We shall add more works on representation learning in the long-tailed setting to the related works section, specifically [5-11]. As for the contributions and distinctness of the paper, most works in long-tailed learning have aimed at creating new methods. We focus on the fundamental scientific phenomena underlying the difficulty of long-tailed learning rather than developing new methods.
>
> 4. The quotations in the abstract are intended to be quotations, and designed to emphasize that the terms used are colloquial and not formal. As for the citation format, we are using the default citation style for ICLR, but we will double check for any inconsistencies.
>
> Thank you for taking the time to read our submission, and I hope this response adequately answers your criticisms.
>
> 1. Liu, W., Wen, Y., Yu, Z., Li, M., Raj, B., & Song, L. (2017). SphereFace: Deep Hypersphere Embedding for Face Recognition. 2017 IEEE Conference on Computer Vision and Pattern Recognition (CVPR), 6738-6746.
> 2. Deng, Jiankang, J. Guo and Stefanos Zafeiriou. “ArcFace: Additive Angular Margin Loss for Deep Face Recognition.” 2019 IEEE/CVF Conference on Computer Vision and Pattern Recognition (CVPR) (2019): 4685-4694.
> 3. Jialun Liu, Yifan Sun, Chuchu Han, Zhaopeng Dou, and Wenhui Li. Deep Representation Learning onLong-Tailed Data: A Learnable Embedding Augmentation Perspective. In2020 IEEE/CVF Conferenceon Computer Vision and Pattern Recognition (CVPR), pages 2967–2976, Seattle, WA, USA, June 2020.IEEE.
> 4. Shuo Yang, Lu Liu, and Min Xu.  FREE LUNCH FOR FEW-SHOT LEARNING: DISTRIBUTIONCALIBRATION. page 13, 2021
> 5. Exploring balanced feature spaces for representation learning. In ICLR, 2021.
> 6. Contrastive learning based hybrid networks for long-tailed image classification. In CVPR, 2021.
> 7. Parametric contrastive learning. In ICCV, 2021.
> 8. Distributional robustness loss for long-tail learning. In ICCV, 2021.
> 9. Yin, Xi, et al. "Feature transfer learning for face recognition with under-represented data." Proceedings of the IEEE/CVF Conference on Computer Vision and Pattern Recognition. 2019.
> 10. Kim, Jaehyung, Jongheon Jeong, and Jinwoo Shin. "M2m: Imbalanced classification via major-to-minor translation." Proceedings of the IEEE/CVF Conference on Computer Vision and Pattern Recognition. 2020.
> 11. Kozerawski, Jedrzej, et al. "BLT: Balancing Long-Tailed Datasets with Adversarially-Perturbed Images." Proceedings of the Asian Conference on Computer Vision. 2020.

---

### Official Review · Reviewer_gifg · 2021-11-03

**Correctness:** 3
**Technical Novelty And Significance:** 1
**Empirical Novelty And Significance:** 2
**Recommendation:** 5
**Confidence:** 5

**Main Review:**

The weakness:
1) The second objective of this paper is to discuss "why data augmentation helps in representation learning". However, in the paper, only positive effects from data augmentation were shown, the reasons and mechanisms were not fully discussed.

2) The overall paper is based on the unserious term "good enough". What is this term defined? How good is good enough? Good enough in terms of what? Generalization and robustness compared to full balanced data sets or in terms of knowledge transfer? If it is the first one, of course, long-tailed representations are less generalized and robust compared to balanced representations. It is not a new idea and is already discussed in [1]. If it is the second one, then the later experiments don't make any sense. And I think when people say long-tail representations are "good enough" in studies like [2], it is more like it is good enough for long-tail learning rather than comparing it to balanced learning.

3) All the experiments seem unfair to me. For example, D_LT are representations from long-tailed data sets, and D* are representations from balanced data sets. Balanced data sets always have much more training samples compared to corresponding long-tailed counterparts. How do you know these inferior results were not caused by the lack of training samples?

4) In the "adding unseen samples" experiments (e.g., Fig 6, Fig 7, Fig 8), only results on D_LT were reported. I want to see results when unseen samples are added to D* as well. Only by doing this can you prove D* is less diffused and better localized.

5) Fig 7 needs a more detailed legend. So many components don't have explanations.

6) By looking at Figure 5, I don't see a significant difference between D_LT and D* in Cifar100-LT and ImageNet-LT.

[1] Liu, Z., Miao, Z., Zhan, X., Wang, J., Gong, B., & Yu, S. X. (2019). Large-scale long-tailed recognition in an open world. In Proceedings of the IEEE/CVF Conference on Computer Vision and Pattern Recognition (pp. 2537-2546).
[2] Kang, B., Xie, S., Rohrbach, M., Yan, Z., Gordo, A., Feng, J., & Kalantidis, Y. (2019). Decoupling representation and classifier for long-tailed recognition. arXiv preprint arXiv:1910.09217.

**Summary Of The Paper:**

This paper tries to prove that there is a bottleneck in feature learning for long-tailed classification and data augmentation can help relieve the issues in long-tail feature space. Three major experiments were done to prove that feature space 1) is more biased than balanced feature space, 2)  is more disused and less compact than balanced feature space, and 3) less localized in terms of feature centroids. And data augmentation can help alleviate all three issues.

**Summary Of The Review:**

I think the intuition of this paper is not clear and the experiments are not persuasive.

---

> ### Author Response · Authors · 2021-11-23
> **Response to weaknesses**
>
> 1. We show that (1) data augmentation improves performance (2) we show that it improves inter-class separation, intra-class compactness, and improves localization of tail classes w.r.t to the true data distribution. This is our proposed explanation for *why* and *how* data augmentation improves performance. As far as we can tell, there are no negative effects of the data augmentation.
>
>
> 2. "Good enough" is a rhetorical device. We do not use the term "good enough" for comparison purposes. We know that representations learned from balanced, complete datasets are better than representations learned from long-tailed datasets. The paper focuses on 3 metrics: inter-class separation, intra-class compactness, and distribution shift of tail classes w.r.t to the true data distribution, and we use those to illustrate the differences between long-tailed representations and representations learned from balanced, complete datasets. Our goal is to show *concrete, quantitative measures* that describe how the long-tailed representations differ from the representations learned from the complete, balanced dataset.
>
> 3. **This is an excellent point and we thank the reviewer for raising it!** We conduct additional experiments for 1.1 to address this concern.
> We have conducted additional experiments by adding a third category of dataset, $\bar{\mathcal{D}}$. In $\bar{\mathcal{D}}$, the number of samples is the same as $\mathcal{D}_{LT}$, with the only difference being that each class has the same number of samples — there are no more few shot classes.
>
> **Additional Experiments with new balanced set**
>
> CIFAR10 | $\bar{\mathcal{D}} \rightarrow \mathcal{D}^\star$ (0.871) >  $\\mathcal{D_{LT}} \rightarrow \mathcal{D}^\star$ (0.826)
>
> CIFAR100 | $\bar{\mathcal{D}} \rightarrow \mathcal{D}^\star$ (0.55) >  $\\mathcal{D_{LT}} \rightarrow \mathcal{D}^\star$ (0.515)
>
> ImageNet | $\bar{\mathcal{D}} \rightarrow \mathcal{D}^\star$ (0.478) >  $\\mathcal{D_{LT}} \rightarrow \mathcal{D}^\star$ (0.458)
>
> In all cases, the representation learned by the balanced dataset **with the name number of samples as the long-tailed dataset** outperforms the representation learned by the long-tailed dataset.
>
> **In all other sections**, the goal is to **illustrate the differences between the representations learned from the complete, balanced data and the representations learned from the long-tailed data**, to reveal *specific, quantitative ways in which the long-tailed representations differ from the ideal representations*. Experiments in sections other than 1.1 intentionally compare the best possible obtainable representations to the long-tailed representations.
>
>
> 4. We have checked Figures 6,7,8 and verified that they show results for $\mathcal{D}^\star$. This is indicated in the titles of each subpanel. We include the locations here for your reference. Figure 6 (bottom right is $\mathcal{D}^\star$). Figure 7 (middle panel is $\mathcal{D}^\star$). Figure 8 (bottom right is $\mathcal{D}^\star$).
>
> 6. The intention of Fig 5 is to show an aspect of the differences between the ideal representation (orange line) and the long-tailed distribution (blue line). It can be seen that the blue line is consistently higher than the orange line for most classes, showing that the variance captured by the first component is higher for the long tailed representation than the ideal representation. This is, to our knowledge, a novel observation. We are showing that one of the differences between long-tailed representations and the ideal representation is that the variance is concentrated in fewer dimensions.
>
> Thank you for your thoughtful comments. We hope our response answers your criticism!

---

### Decision · Program_Chairs · 2022-01-20

**Decision:**

Reject

**Comment:**

This paper investigates the role of representation learning when the distribution over the feature space has a long tail. The main motivation is to determine how much of the overall learning, in this case, is bottlenecked specifically by representation learning. The main findings are that vanilla learning gives brittle long-tailed representations, harming overall performance. The paper suggests a form of data augmentation to remedy this. Reviewers acknowledge that this investigation is worthwhile. However, many concerns were raised as to whether experiments support the drawn conclusions. A more principled approach to the data augmentation methodology is also needed. The authors address some of these, providing further experiments, but these were not enough to sway reviewers. Since results are fundamentally empirical in nature, this shortcoming indicates that the paper is not ready to share with the community just yet. Stronger experiments with clearer evidence are needed to fully support the thesis of the work.